# STATED CAUSAL LANGUAGE MODELING: OFF-THE-SHELF ENHANCEMENT OF CONTEXT MEMORIZATION

## ABSTRACT

We propose stated causal language modeling (stated-CLM), a novel method to enhance the memory capacity of large language models (LLMs) without modifying their architecture or parameters. Unlike existing context segmentation and sliding methods that discard low-weight tokens, stated-CLM compresses adjacent tokens, significantly reducing context information loss. We utilize the classic network pruning techniques with second-order derivatives to optimize the compressed token in the differentiable key-value space. Experiments on LLaMA, Mistral, and Gemma demonstrate that stated-CLM outperforms baselines on the LongBench benchmark by an average of 6.12% (LLaMA3.1-8B) and 5.97% (Mistral-v0.3-7B). On TopicRet, stated-CLM achieves accuracy levels comparable to full context models, while the baselines' accuracy is close to zero.

## 1 INTRODUCTION

Memorizing certain knowledge is a critical requirement for large-scale language models (LLMs). For example, a company's enterprise chatbot needs to memorize necessary facts about its products. Existing LLMs with sequence-to-sequence and attention mechanisms have inherent memorization limitations: they rely on context to represent memory, meaning that this relevant knowledge must be included within the context. This approach faces two main challenges: a rigid limitation on context length and the quadratic growth in computational complexity.

Our goal is to enable LLMs to remember more without excessively occupying context length. A typical approach is to add a differentiable external memory module to existing models (Weston et al., 2015)(Wu et al., 2022)(Munkhdalai et al., 2024)(He et al., 2024)(Tworkowski et al., 2024). The additional memory module encodes contexts, enhancing the memorization ability by retrieving the module when predicting the next token. However, this memory enhancement requires architecture modification and adaptive fine-tuning of the original LLMs, which increases the additional training overhead. The fine-tuning data may be biased and compromise the generalizability of the original model.

To retain the generalization abilities of LLMs, we aim to enhance the memory capacity of original LLMs without architecture or parameter modification. This means that the same length of contextual tokens need to memorize more information for upcoming token predictions. Recently emerging **context segmentation and sliding** methods have been exploring this route, including StreamingLLM (Xiao et al., 2023), LazyLLM (Fu et al., 2024), and LongCache (Liu et al., 2024b). The core strategy of these methods is to retain only the critical tokens in the context (e.g., tokens with the highest attention scores) and discard other tokens. Although this route does not disrupt the original LLMs, the context is compromised. A large number of low-weight tokens that may still have an impact on the token predictions are directly discarded, resulting in context information loss.

In this paper, we compress rather than discard low-weight tokens while keeping the architecture and parameters in the original LLMs. This reduces context information loss while inheriting the generalization ability of the original LLMs. Our intuition lies in the fact that language can naturally be shortened. A simple example is that `Alice went to the library. The library was quiet.` can be shortened to `Alice went to the library.` **It** `was quiet.` LLMs have the ability to make consistent subsequent predictions for both contexts. From a memory perspective, LLMs have the ability to memorize more information with fewer tokens (i.e., using "It" to memorize "The library").

This inspires us to propose the stated causal language modeling (stated-CLM), aimed at enhancing the context-memorization capabilities of LLMs. We extend the iterative next token prediction in the existing CLM and propose an iterative next token prediction and adjacent token compression method. The core idea relies on a simple yet powerful operation - adjacent token compression — to achieve atomic memorization operations. More specifically, after generating each next token, we select a pair of adjacent tokens in the context and compress them into one token, thereby maintaining the context at a desired length. Compared to existing context segmentation methods, the compressed token inherits the information of two original tokens, thus reducing information loss.

We compare stated-CLM with context segmentation methods in Figure 1. StreamingLLM and Long-Cache discard a large number of low-weight tokens, resulting in a loss of context information. In contrast, stated-CLM preserves context information to the greatest extent possible while reducing token volumes. Furthermore, since each atomic compression operation only involves two adjacent tokens, stated-CLM incurs lower compression loss per iteration compared to traditional blockwise encoding and compression methods (Xiao et al., 2024)(Munkhdalai et al., 2024). This makes it more feasible for the iterative token-by-token generation process. The stated-CLM can be adapted to existing CLMs without any fine-tuning.

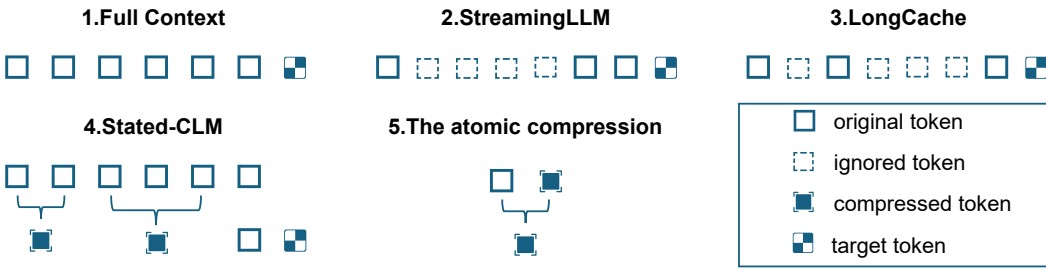

Figure 1: Comparison with context segmentation methods. Stated-CLM achieves memory of long contexts and reduces information loss through iterative compressions. Each atomic compression compresses two adjacent tokens into one token.

To implement adjacent token compression without fine-tuning model parameters, our goal is to find an optimal replacement token for a pair of tokens (for example, using "It" to replace "The library"). First, to make the optimization differentiable, we perform compression in the key-value space instead of the token space. We refer to network pruning techniques (LeCun et al., 1989), using second derivatives to optimize and compute the optimal compressed token encoding, and employ the Fisher Information Matrix to enhance computational efficiency. We also ensure that in the attention mechanism, once the key-values are replaced with multiple identical tokens, any new query can produce consistent outputs. To this end, we extend the original attention mechanism to accurately restore the outputs before compression.

We conducted extensive experiments over LLaMA (Dubey et al., 2024), Mistral, and Gemma (Team et al., 2024). The results demonstrate that stated-CLM significantly enhances the memory capacity of the original LLMs. On the LongBench benchmark, the stated-CLM based on LLaMA3.1-8B outperformed the competitors by an average of 6.12%, while the Mistral-v0.3-7B-based version showed an average improvement of 5.97%. In the historical information retrieval task TopicRet, where competitors' accuracy was nearly zero, stated-CLM achieved accuracy levels comparable to those obtained using full context.

## 2 RELATED WORK

Enhancing the memorization capabilities of language models has emerged as a critical problem. Various methods have been proposed for long-context LLMs. We discuss three techniques that are more related to our study below.

**Sparse Attention Mechanisms** The original Transformer has $O(n^2)$ complexity for a sequence with $n$ tokens, due to the dense attention access. Therefore, a simple way is to reduce the density of attention to $O(n)$. Typical approaches include *Sparse Transformer* (Child et al., 2019), *Long-*

*former* (Beltagy et al., 2020), and *Big Bird* (Zaheer et al., 2020). These methods leverage some prior dependency assumptions to optimize the attention patterns, significantly reducing computational complexity and enabling the model to handle longer sequences. However, such prior assumptions may not always hold in practice, and the resulting sparsity may not be effective in capturing all relevant dependencies. Besides, adapting existing LLMs with dense attention to sparse attention is nontrivial and leads to generalization capability loss.

**External Memory Modules** To expand the memory capacity of models, researchers have incorporated external knowledge storage modules (Weston et al., 2015)(Wu et al., 2022)(Munkhdalai et al., 2024)(He et al., 2024)(Tworkowski et al., 2024). *Memorizing Transformers* shows that related key-value pairs can be retrieved through a kNN lookup. *Focused Transformer* (Tworkowski et al., 2024) uses contrastive learning to help identify relevant and irrelevant tokens. *LongMem* (Wang et al., 2024) and *CAMELoT* (He et al., 2024) uses decoupled LLMs and memory modules. During training the memory modules, the parameters of the original LLMs are frozen. Although additional memory modules can compress context, adjustments to the model architecture require retraining and biases in the training data will be introduced into the model.

**Context Segmentation and Sliding** Researchers have developed various context strategies that optimize the contexts for memorization, instead of optimizing the models (Ge et al., 2024) (Mu et al., 2024). Typical methods include *StreamingLLM* (Xiao et al., 2023), *LazyLLM* (Fu et al., 2024), and *LongCache* (Liu et al., 2024b). These strategies dynamically select and retain key tokens, thereby reducing the computational and storage burden when processing long sequences. *StreamingLLM* ensures that the model focuses on the initial (sink) tokens and the recent tokens. *LazyLLM* employs a dynamic pruning strategy that selects different subsets of tokens in different generation steps, although they might be pruned in previous steps. *LongCache* intelligently manages the cache by expanding StreamingLLM to keep critical middle tokens. However, these compression strategies essentially involve directly discarding tokens with lower weights, which results in significant information loss during the process.

## 3 METHODS

### 3.1 MEMORIZING ON NATIVE CASUAL LANGUAGE MODELING

Traditional causal language modeling (CLM) adopts the next-word prediction: iteratively predicting the next token based on historical tokens. This paradigm demonstrates strong inference capabilities, but has limitations in memory. It is formulated below:

$$P_{CLM}(w_{n+1} \mid w_1, \ldots, w_n) \tag{1}$$

where $w_{n+1}$ denotes the current token, and $w_1, \ldots, w_n$ represent the historical tokens. As we stated in the introduction, the memorization capacity of CLMs is limited by the number of tokens $n$.

To enhance the memorization, we introduce the **stated causal language modeling** (stated-CLM) on the native CLM. In stated-CLM, each token can be either an original input token or represent the memory of a span of continuous tokens. To keep a desired number of tokens in the context, the next token prediction is expanded to token generation and memorization.

**Generation**-stage: Predicts the next token using existing tokens:

$$P_{CLM}(w_{n+1} \mid \tilde{w}_1, \tilde{w}_2, \ldots, \tilde{w}_n) \tag{2}$$

where $\tilde{w}_i$ represents a mixed token, which can be either an original token or reprsenting the memory of a context span. Note that we do not manuplate the probabilistic model $P_{CLM}$, the existing CLM architecture and its pre-trained parameters are retained to pursue model generalization ability.

**Memorization**-stage: Compresses the long context into a shorter one while maintaining the consistency of the subsequent predictions. This is achieved by an atomic operation: compress an adjacent token pair from the context into a single token, so that the context length is reduced from $n+1$ to $n$:

$$\text{compress}(e_m, e_{m+1}) : \mathbb{R}^{2 \times d} \to \mathbb{R}^d \tag{3}$$

Here, $e_m$ and $e_{m+1}$ are the embeddings of adjacent tokens $w_m$ and $w_{m+1}$, respectively. compress($e_m, e_{m+1}$) encodes these two adjacent tokens into a single token, thus achieving basic context compression functionality.

**For example:** We compare the original paradigm and the new paradigm as follows. In the original paradigm, inference needs to store and encode all seven input tokens. In the new paradigm, inference only needs to store and encode four tokens after three compressions (each compression is marked by a │ red frame │).

- **CLM** uses all seven tokens to predict the next token:

$$P\left(w_t \mid \text{The, quick, brown, fox, jumps, over, the}\right) \tag{4}$$

- **Stated-CLM** uses four original/compressed tokens to predict the next token:

$$P\left(w_t \mid \boxed{\text{compress(The, quick)}}, \text{brown, fox}, \boxed{\text{compress(}\boxed{\text{compress(jumps, over)}}, \text{the)}}\right) \tag{5}$$

In the new paradigm, input tokens can be original tokens, compressed tokens (e.g. compress(jumps, over)), or nested compressed tokens (e.g., compress(compress(jumps, over), the)). Thus, the compression can be carried out multiple times to always keep a desired number of tokens to store and encode.

We highlight that atomic operation $\text{compress}(w_m, w_{m+1})$ offers advanced features for context memorization by inheriting the original CLM architecture. The compressed tokens remain compatible with the CLM's mechanism. As a result, it significantly enhances the LLM's memorization ability without additional fine-tuning and architecture modification, thereby preserving its generalization ability.

A key aspect of an effective context compression storage method lies in designing compression strategies to maintain prediction consistency before and after compression. We will elaborate on this in § 3.2.

## 3.2 OPTIMAL TOKEN COMPRESSION

Given the embedding vectors of the original tokens $e_1, e_2, \ldots, e_n$, and one pair of adjacent positions $m, m+1$ to be compressed, the CLM loss function before compression is $\text{Loss}_{\text{CLM}}(e_1, e_2, \ldots, e_n)$. After compression, the loss function becomes:

$$\text{Loss}_{\text{CLM}}\left(e_1, \ldots, e_{m-1}, \text{compress}(e_m, e_{m+1}), e_{m+2}, \ldots, e_n\right) \tag{6}$$

Our objective is to find the optimal compressed embedding that minimizes the disturbance.

In particular, compression changes the input dimensionality, reducing the input vector dimensions from $n \times d$ to $(n-1) \times d$. This makes directly computing the optimal compression challenging. Therefore, we the optimal compressed token embedding via the following two steps:

1. **Convert Adjacent Tokens into Identical Embeddings.** Replace $(e_m, e_{m+1})$ with two identical vectors $(e^*, e^*)$, thus maintaining the dimensionality of the input.

2. **Expand the Attention Mechanism to Support Identical Embeddings.** Expand the attention mechanism to efficiently handle tokens with multiple identical embeddings. The outputs need to be consistent with the uncompressed tokens for any query embeddings.

### 3.2.1 FIRST STEP: CONVERT ADJACENT TOKENS INTO IDENTICAL EMBEDDINGS

We consider a simplified setting where two identical embeddings $e, e$ replace the original embeddings $e_m, e_{m+1}$ without reducing the number of tokens. Since tokens other than $e_m, e_{m+1}$ remain unchanged, we can focus on minimizing the loss function with respect to these two tokens. To simplify notation, we represent the loss function as follows:

$$\mathcal{L}(e_m, e_{m+1}) = \text{Loss}_{\text{CLM}}\left(e_1, \ldots, e_{m-1}, e_m, e_{m+1}, e_{m+2}, \ldots, e_n\right) \tag{7}$$

We will demonstrate that leveraging the properties of the attention mechanism allows for lossless adaptation in the second step (3.2.2). Therefore, minimizing the loss after compression is equivalent to finding the optimal $e^*$ in the first step.

$$e^* = \arg\min_e \mathcal{L}(e, e) \tag{8}$$

Performing a second-order Taylor expansion of $\mathcal{L}(x, y)$ around $(e_m, e_{m+1})$:

$$\mathcal{L}(x, y) \approx \mathcal{L}(e_m, e_{m+1}) + \nabla\mathcal{L}(e_m, e_{m+1})^\top \begin{pmatrix} x - e_m \\ y - e_{m+1} \end{pmatrix} + \frac{1}{2} \begin{pmatrix} x - e_m \\ y - e_{m+1} \end{pmatrix}^\top H \begin{pmatrix} x - e_m \\ y - e_{m+1} \end{pmatrix} + \mathcal{O}(\Delta^3) \tag{9}$$

where $H$ is the Hessian matrix at $(e_m, e_{m+1})$, represented as:

$$H = \begin{pmatrix} H^{11} & H^{12} \\ H^{21} & H^{22} \end{pmatrix} \tag{10}$$

Each submatrix $H^{ab}$ is a $d \times d$ matrix. To minimize $\mathcal{L}(e, e)$, we set $x = e, \quad y = e$:

$$\mathcal{L}(e, e) \approx \mathcal{L}(e_m, e_{m+1}) + \nabla\mathcal{L}(e_m, e_{m+1})^\top \begin{pmatrix} e - e_m \\ e - e_{m+1} \end{pmatrix} + \frac{1}{2} \begin{pmatrix} e - e_m \\ e - e_{m+1} \end{pmatrix}^\top H \begin{pmatrix} e - e_m \\ e - e_{m+1} \end{pmatrix} \tag{11}$$

Defining the gradient vector as follows:

$$\nabla\mathcal{L}(e_m, e_{m+1}) = \begin{pmatrix} g_m \\ g_{m+1} \end{pmatrix} \tag{12}$$

where $g_m = \nabla_{e_m}\mathcal{L}$ and $g_{m+1} = \nabla_{e_{m+1}}\mathcal{L}$.

Setting the derivative to zero in equation 11 yields the optimal solution below. More details are shown in Appendix A.

$$e^* = \left(H^{11} + 2H^{12} + H^{22}\right)^{-1} \left(H^{11}e_m + H^{12}(e_m + e_{m+1}) + H^{22}e_{m+1} - (g_m + g_{m+1})\right) \tag{13}$$

To compute the Hessian matrix $H$, we utilize the Fisher information matrix. We follow the network pruning technique (LeCun et al., 1989; Kim et al.; Cui & Wang, 2024) to assume that the cross-parameter interactions are neglibible and approximate the Fisher information matrix as a diagonal matrix. The diagonal elements can be efficiently computed using their gradients:

$$H_{ii} = F_{ii} = \nabla\mathcal{L}(e_m, e_{m+1})_i^2 \tag{14}$$

One issue with the above diagonal matrix approximation is that it leads to a reduction in the scale of quadratic terms relative to gradient terms in equation 13. To address this, considering that CLMs have seen samples similar to the current input during large-scale pre-training, their gradients $g_m, g_{m+1}$ tend to be zero. Therefore, we ignore the gradient terms in equation 13.

In practical computations, each $e_i$ contains both a key and a value. For $e_m, e_{m+1}$, let the key vectors be denoted as $K_m$ and $K_{m+1}$, and the value vectors as $V_m$ and $V_{m+1}$. We replace $(K_m, K_{m+1})$ with $(K_m^*, K_m^*)$, and $(V_m, V_{m+1})$ with $(V_m^*, V_m^*)$.

### 3.2.2 Second Step: Lossless Representing Identical Embeddings in Attention

In § 3.2.1, we have already replaced the embeddings of the two tokens with identical embeddings. We now show how to efficiently compute the attention with the identical embeddings, so that the output remains consistent for any query.

In the original attention mechanism, the output for the query $q$ is:

$$\text{output}(q) = \text{Attention}(q, K, V) = \sum_{i=1}^{n} \frac{\exp(q \cdot K_i)}{\sum_{j=1}^{n} \exp(q \cdot K_j)} V_i \tag{15}$$

After compression, since $K_m = K_{m+1} = K_m^*$ and $V_m = V_{m+1} = V_m^*$, we optimize embedding storage by reorganizing the key and value vectors as:

$$K'_1, K'_2, \ldots, K'_{n-1} = K_1, \ldots, K_m, K_{m+2}, \ldots, K_n$$
$$V'_1, V'_2, \ldots, V'_{n-1} = V_1, \ldots, V_m, V_{m+2}, \ldots, V_n \quad (16)$$

Thus, the computation of $\text{output}(q)$ can be simplified to:

$$\text{output}(q) = \sum_{i=1}^{n-1} \text{cnt}_i \frac{\exp(q \cdot K'_i)}{\sum_{j=1}^{n-1} \exp(q \cdot K'_j)} V'_i \quad (17)$$

where $\text{cnt}_i = \begin{cases} 1 & \text{if } i \neq m \\ 2 & \text{if } i = m \end{cases}$, indicating how many original tokens correspond to the $i$-th token. This allows us to process only $n-1$ tokens, achieving both storage and computational efficiency.

The above method can be extended to arbitrary iterations of compressions. Specifically, we maintain $\text{cnt}_i$ to represent the number of original tokens correspond to the current $i$-th token. Initially, $\text{cnt}_i = 1$ for all original tokens. Each time token $m$ and $m+1$ are compressed, we set $\text{cnt}_m \leftarrow \text{cnt}_m + \text{cnt}_{m+1}$. Compared to the original attention mechanism, we only need to maintain $O(n)$ additional integers.

### 3.3 COMPRESSION POSITION SELECTION

We explain how to select the compression position $m$. To minimize information loss and maintain the model's predictive capability, we adopt the position with least affect in the attention.

Specifically, we use the attention mechanism to calculate the importance score for each token. For each token $w_i$, we compute the sum of attention weights from different queries and then sum the scores across different attention heads to obtain the complete score of the token. We choose the adjacent pair of tokens $m, m+1$ with the lowest attention scores.

$$\text{score}_i = \sum_{h=1}^{H} \sum_{j=1}^{n-1} \text{Attention}_{j,i}^{(h)} + \text{Attention}_{j,i+1}^{(h)} \quad (18)$$

where $H$ denotes the number of attention heads, and $\text{Attention}_{j,i}^{(h)}$ represents the attention weight from query $Q_j$ to key $K_i$ in the $h$-th attention head.

In the Transformer model, we compute the token scores at each layer and select the token with the lowest score separately. Different positions $m$ are chosen in different layers, thereby improving the flexibility to minimize the information loss for compression, maximizing the preservation of the model's performance.

**Positional Normalization** In practice, we observed that the aforementioned *score* tends to assign lower scores to earlier tokens. To address this, we further normalize *score* to better align with our expectations. Specifically, we employ:

$$\text{score}'_i = \exp(-i/\sigma)\text{score}_i \quad (19)$$

Furthermore, based on the characteristic of sink tokens identified in previous work (Xiao et al., 2023), we define $\text{score}'_i = \inf$ for the first `sink_length` tokens, ensuring that these tokens are always retained in the context. In this paper, we set $\sigma$ to 4096 and `sink_length` to 32.

Combining with § 3.2.1 and § 3.3, we can obtain an efficient compression algorithm. This algorithm keeps a desired number of tokens by continuously adding new tokens into it while minimizing the model loss.

### 3.4 EFFICIENT CHUNKING-WISE COMPRESSION

To enhance the efficiency of stated-CLM, we adopt a chunking-wise compression instead of compression after each new token. We generate tokens one by one as in the original CLM decoding until the context length reaches `max_context_length`. Then after every $k$ tokens, we compress $k$ tokens at once. To achieve this, we calculate the $k$ pairs of adjacent tokens with the lowest scores according to equation 19, and compute the optimal compression for these tokens according to equation 13.

This approach allows compressing $k$ tokens through a single forward and backward. As a result, the model maintains almost the same efficiency as in the original CLM decoding.

# 4 EXPERIMENTS

## 4.1 SETUP

We aim to verify the effect of stated-CLM in memorizing long contexts. To achieve this, we compare with the following baselines:

- **Full Contexts Inference in Standard CLM**: Serves as the upper bound for performance but requires substantial computational resources.

- **Previous Context Segmentation Methods**: These are previous methods which also leverage the native LLMs and shorten the contexts. In the experiments, we compare with two typical baselines: StreamingLLM Xiao et al. (2023) and LongCache Liu et al. (2024b).

We selected several advanced CLMs as the base models for our experiments, including LLaMA2-7B-chat (Touvron et al., 2023), LLaMA3-8B-instruct (Dubey et al., 2024), LLaMA3.1-8B-instruct, Mistral-7B-Instruct-v0.3, Gemma-1.1-7B-it (Team et al., 2024), and Gemma-1.1-2b-it. When adapting to stated-CLM, we do not make any architectural adjustments or fine-tuning. We use different context length limits (i.e. cache size for StreamingLLM and LongCache) for different models. More details are shown in Appendix B.

All experiemnts run on a single NVIDIA A100 80 GiB GPU.

## 4.2 LONG CONTEXT PERFORMANCE EVALUATION

We evaluate the performance of stated-CLM in long-context QA tasks to validate its ability to memory contexts. The experiments are conducted using LongBench (Bai et al., 2023), a benchmark specifically designed for long-context scenarios. Given that stated-CLM supports unlimited context lengths, we do not truncate the input data in our experiments except for the full context baseline.

**Results** We present the experimental results in Table 1. It can be seen that stated-CLM outperforms StreamingLLM and LongCache across all models. On LLaMA3-8B, LLaMA3.1-8B, and Mistral-v0.3-7B, stated-CLM significantly improves over LongCache by more than 6.10, 6.12, and 5.97, respectively. This demonstrates that the token compression used by stated-CLM has a significant advantage over its competitors.

**Ability to Extend Model Context Limits** We noticed that for the two models with originally limited context lengths, LLaMA2-7B and LLaMA3-8B, the performance of stated-CLM is close to or exceeds that of using the full context. This shows that stated-CLM can remember long contexts without being limited by the original maximum context length of the models.

## 4.3 CAN THE MODEL STILL REMEMBER EARLY HISTORICAL CONTEXT?

We examine whether the model can retrieve information over lengthy contexts. Previous research has shown that models tend to forget earlier (but not necessarily the earliest) parts of the context in long sequences (Li et al., 2023)(Liu et al., 2024a). Notably, when querying about the second or third topic in a long text, previous literature has shown that many open-source models' accuracy approaches zero (An et al., 2023). Consequently, TopicRet (Li et al., 2023) was proposed to evaluate models' ability to remember information from earlier parts of the context. We utilize the version from L-Eval (An et al., 2023), as its questions are more challenging and demand higher performance.

The results are presented in Table 2. It is evident that previous context segmentation methods perform poorly on this task, with effectiveness close to zero. This is because the baselines have already discarded tokens from earlier positions, resulting in the context no longer containing information about the original second or third topic. In contrast, stated-CLM demonstrates significant performance improvements, achieving results comparable to or even surpassing those of full context models.

Table 1: Performance on LongBench. Stated-CLM outperforms its baselines on all models.

| | Single-Doc QA | Multi-Doc QA | Sum | Few-shot Learning | Synthetic | Code | Avg. |
|---|---|---|---|---|---|---|---|
| **LLaMA2-7B** | | | | | | | |
| Full Context | 20.46 | 17.46 | 18.46 | 47.97 | 5.32 | 54.31 | 27.33 |
| - StreamingLLM | 16.10 | 16.34 | 17.49 | 49.61 | 2.73 | 45.71 | 24.66 |
| - LongCache | 16.27 | 16.29 | 17.58 | 49.51 | 2.62 | **46.21** | 24.74 |
| **- Stated-CLM** | **19.98** | **18.52** | **18.47** | **50.12** | **4.60** | 45.76 | **26.24** |
| **LLaMA3-8B** | | | | | | | |
| Full Context | 35.15 | 30.39 | 20.98 | 56.75 | 33.61 | 52.99 | 38.31 |
| - StreamingLLM | 35.56 | 25.01 | 19.25 | 56.27 | 35.94 | 49.89 | 36.98 |
| - LongCache | 36.04 | 25.27 | 19.23 | 56.06 | 38.29 | 50.00 | 37.48 |
| **- Stated-CLM** | **40.05** | **29.91** | **19.53** | **57.39** | **60.17** | **54.46** | **43.58** |
| **LLaMA3.1-8B** | | | | | | | |
| Full Context | 41.52 | 28.79 | 25.23 | 60.64 | 58.05 | 58.92 | 45.52 |
| - StreamingLLM | 37.66 | 29.36 | 24.01 | 58.20 | 40.93 | 53.92 | 40.68 |
| - LongCache | 38.26 | 29.90 | 24.07 | 58.76 | 44.88 | **54.11** | 41.66 |
| **- Stated-CLM** | **46.05** | **39.18** | **25.45** | **58.77** | **64.29** | 52.94 | **47.78** |
| **Mistral-v0.3-7B** | | | | | | | |
| Full Context | 38.49 | 35.33 | 25.07 | 60.44 | 51.90 | 59.19 | 45.07 |
| - StreamingLLM | 32.57 | 25.86 | 23.74 | 56.72 | 31.78 | **54.55** | 37.54 |
| - LongCache | 32.90 | 26.09 | 23.80 | **57.50** | 34.86 | 54.28 | 38.24 |
| **- Stated-CLM** | **38.32** | **33.17** | **24.98** | 57.35 | **57.17** | 54.29 | **44.21** |
| **Gemma-1.1-7B** | | | | | | | |
| Full Context | 36.48 | 23.37 | 22.10 | 51.30 | 45.05 | 46.04 | 37.39 |
| - StreamingLLM | 32.67 | 18.11 | 21.39 | 51.13 | 34.44 | 41.96 | 33.28 |
| - LongCache | 33.35 | 17.97 | 21.50 | 51.80 | 34.29 | **41.99** | 33.48 |
| **- Stated-CLM** | **40.85** | **27.29** | **22.59** | **53.18** | **40.59** | 41.52 | **37.67** |
| **Gemma-1.1-2B** | | | | | | | |
| Full Context | 28.50 | 16.84 | 20.85 | 45.19 | 3.76 | 46.11 | 26.87 |
| - StreamingLLM | 24.16 | 16.60 | 18.69 | 40.22 | 3.89 | 35.36 | 23.15 |
| - LongCache | 26.11 | 16.57 | 19.38 | **44.99** | 4.04 | **36.44** | 24.59 |
| **- Stated-CLM** | **27.91** | **18.41** | **19.82** | 42.96 | **4.25** | 36.26 | **24.93** |

Table 2: Model Performance on early topic retrieval. L. denotes LLaMA, M. denotes Mistral, G. denotes Gemma. Stated-CLM has comparable performance with full context models, while previous context segmentation methods perform poorly.

| Model | L.2-7B | L.3-8B | L.3.1-8B | M.7B | G.7B | G.2B |
|---|---|---|---|---|---|---|
| Full Context | 73.33 | 76.67 | 80 | 42.67 | 24.67 | 20.67 |
| - StreamingLLM | 0.00 | 0.00 | 0.00 | 0.00 | 0.00 | 0.00 |
| - LongCache | 0.00 | 0.00 | 0.00 | 8.67 | 0.67 | 0.00 |
| **- Stated-CLM** | **28.00** | **64.00** | **76.00** | **35.33** | **46.67** | **32.00** |

## 4.4 COMPRESSION RATE ANALYSIS

We directly evaluate the information loss caused by stated-CLM's context compression. To this end, we measure the performance of stated-CLM at different compression rates. The compression rate is defined as the ratio of the number of tokens after compression to the number of tokens before compression. We use the HotpotQA (Yang et al., 2018) task from LongBench for this evaluation.

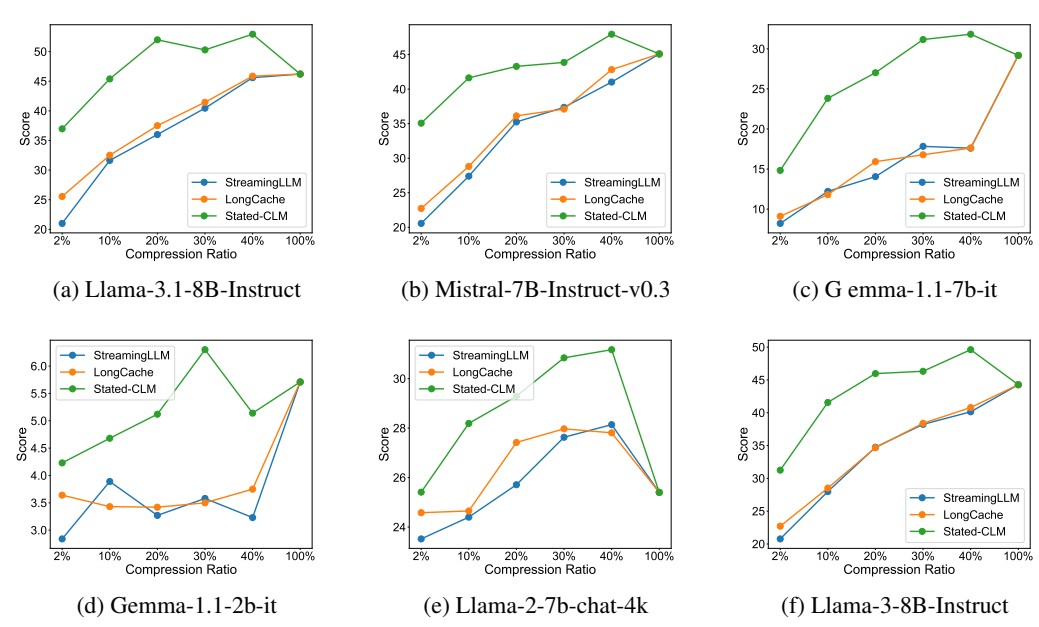

(a) Llama-3.1-8B-Instruct  (b) Mistral-7B-Instruct-v0.3  (c) G emma-1.1-7b-it

(d) Gemma-1.1-2b-it  (e) Llama-2-7b-chat-4k  (f) Llama-3-8B-Instruct

Figure 2: Effect of different compression ratios. Even when compressed to only 10% of the original context length, the model can still maintain performance close to that of the full context.

The results are shown in Figure 2. Stated-CLM exhibits the best information compression capabilities. In most cases, even when compressed to only 10% of the original context length, the model can still maintain performance close to that of the full context. Compared to the baselines, it shows a significant advantage in minimizing information loss. At a compression ratio of 40%, the performance of stated-CLM even surpasses that of using full contexts (100%). This indicates that stated-CLM can effectively filter tokens, thereby reducing the difficulty of model inference.

## 4.5 INFERENCE EFFICIENCY ANALYSIS

We analyzed the inference efficiency of different models. To do this, we had the models generate text using greedy sampling and measured the time required to generate different numbers of tokens. During the generation process, we ignored the context length limitations of the base models.

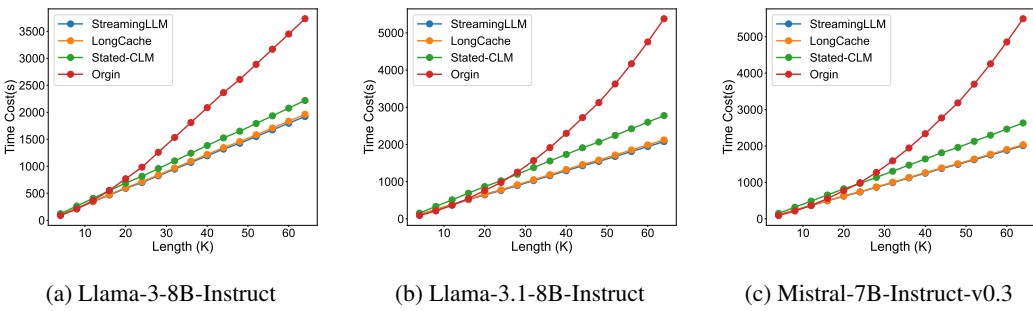

(a) Llama-3-8B-Instruct  (b) Llama-3.1-8B-Instruct  (c) Mistral-7B-Instruct-v0.3

Figure 3: Inference efficiency of different models.

The results can be seen in Figure 3. The inference time for compression methods shows a linear relationship with length, while the original full context inference time exhibits a quadratic relationship.

## 5 CONCLUSION

In this paper, we introduced stated causal language modeling, a novel approach to enhance the context memorization capabilities of large language models without modifying their architecture or parameters. By leveraging the natural compressibility of language, our method compresses adjacent tokens in the context rather than discarding them, significantly reducing information loss compared to existing context segmentation methods.

Our experiments across multiple LLM architectures, including LLaMA, Mistral, and Gemma, demonstrate the effectiveness of stated-CLM in improving memory capacity. On the LongBench benchmark, stated-CLM outperformed baselines by an average of 6.12% with LLaMA3.1-8B and 5.97% with Mistral-v0.3-7B. In the challenging historical information retrieval task TopicRet, stated-CLM achieved accuracy levels comparable to full context use, where competitors' performance was near zero.

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

## A  SOLVING FOR THE OPTIMAL EMBEDDING VECTOR

Continue from equation 11:

**Gradient Term Expansion:**

$$\nabla\mathcal{L}(e_m, e_{m+1})^\top \begin{pmatrix} e - e_m \\ e - e_{m+1} \end{pmatrix} = g_m^\top(e - e_m) + g_{m+1}^\top(e - e_{m+1}) \tag{20}$$

**Quadratic Term Expansion:**

$$\frac{1}{2} \begin{pmatrix} e - e_m \\ e - e_{m+1} \end{pmatrix}^\top H \begin{pmatrix} e - e_m \\ e - e_{m+1} \end{pmatrix} = \frac{1}{2}(e - e_m)^\top H^{11}(e - e_m) + (e - e_m)^\top H^{12}(e - e_{m+1})$$
$$+ \frac{1}{2}(e - e_{m+1})^\top H^{22}(e - e_{m+1}) \tag{21}$$

**Constructing the Total Objective Function**

Adding the above terms, the objective function with respect to $e$ is:

$$\mathcal{L}(e) \approx \mathcal{L}(e_m, e_{m+1}) + g_m^\top(e - e_m) + g_{m+1}^\top(e - e_{m+1})$$
$$+ \frac{1}{2}(e - e_m)^\top H^{11}(e - e_m) + (e - e_m)^\top H^{12}(e - e_{m+1}) + \frac{1}{2}(e - e_{m+1})^\top H^{22}(e - e_{m+1}) \tag{22}$$

**Taking the Derivative of $\mathcal{L}(e)$ with Respect to $e$ and Setting to Zero**

Taking the derivative:

$$\frac{\partial \mathcal{L}}{\partial e} = g_m + g_{m+1} + H^{11}(e - e_m) + H^{12}(e - e_{m+1}) + H^{12\top}(e - e_m) + H^{22}(e - e_{m+1}) \tag{23}$$

Since the Hessian matrix is symmetric, i.e., $H^{12} = H^{21\top}$, we have:

$$\frac{\partial \mathcal{L}}{\partial e} = g_m + g_{m+1} + \left(H^{11} + 2H^{12} + H^{22}\right) e - \left(H^{11}e_m + H^{12}(e_m + e_{m+1}) + H^{22}e_{m+1}\right) \tag{24}$$

Setting the derivative to zero yields the optimal condition:

$$\left(H^{11} + 2H^{12} + H^{22}\right) e^* = H^{11}e_m + H^{12}(e_m + e_{m+1}) + H^{22}e_{m+1} - (g_m + g_{m+1}) \tag{25}$$

**Solving for the Optimal Embedding Vector $e^*$**

The optimal embedding vector $e^*$ is obtained as:

$$e^* = \left(H^{11} + 2H^{12} + H^{22}\right)^{-1} \left(H^{11}e_m + H^{12}(e_m + e_{m+1}) + H^{22}e_{m+1} - (g_m + g_{m+1})\right) \tag{26}$$

# B    MODEL DETAILS

The specific configurations of `sink_length`, `max_context_length` and `chunk_size` as in Table 3. StreamingLLM and LongCache use the same hyper-parameters.

Furthermore, StreamingLLM and LongCache require additional top-k and top-k' parameters. We adopt the recommended settings from LongCache, where top-k is uniformly set to 4, and top-k' varies depending on the model: for LLaMA2-7b-Chat, the top-k' value is set to 48, while for LLaMA3-8b-Instruct , and LLaMA3.1-8b-Instruct and Gemma1.1-2B-Instruct, it is set to 96. For other models, the top-k' value is set to 256. These parameter settings aim to maximize model performance while maintaining consistency and comparability across experiments.

Table 3: Core Parameter Settings for StreamingLLM and LongCache

| Model | Sink Length | Max Content Length | Chunk Length |
|---|---|---|---|
| LLaMA2-7B-Chat | 32 | 2048 | 512 |
| LLaMA3-8B-Instruct | 32 | 4096 | 512 |
| LLaMA3.1-8B-Instruct | 32 | 4096 | 512 |
| Gemma1.1-7B-Instruct | 32 | 8192 | 512 |
| Gemma1.1-2B-Instruct | 32 | 4096 | 512 |
| Mistral-7B-Instruct-v0.3 | 32 | 8192 | 512 |

