# OpenReview forum: "Stated Causal Language Modeling: Off-the-Shelf Enhancement of Context Memorization"
_ICLR.cc/2025/Conference — Submitted to ICLR 2025_

### Official Review · Reviewer_7BNf · 2024-10-30

**Soundness:** 3
**Presentation:** 4
**Contribution:** 3
**Rating:** 5
**Confidence:** 4

**Summary:**

This work introduces a novel training-free approach, Stated-CLM, to enhance the LLMs' capability to memorize contextual information and process long input sequences. The approach compresses the key and value representations of adjacent tokens that receive less attention. Optimal compressed representations, which maintain prediction consistency before and after compression, are calculated using second derivatives under certain assumptions inspired by prior network pruning techniques. Experiments conducted on both LongBench and TopicRet demonstrate the favorable performance of Stated-CLM over StreamingLLM and LongCache across several backbone LLMs.

**Strengths:**

1. The paper is clearly written and easy to follow.
2. The proposed approach is well-motivated, and supported by theoretical analysis based on specific assumptions.
3. The author did extensive experiments on six LLMs to validate the effectiveness of the proposed approach. The analysis indicates the superior performance of Stated-CLM over baselines under various compression ratios and shows the linear relationship between time cost and context length of compression methods.

**Weaknesses:**

The baselines utilized in this work are relatively weak. More competitive baselines, such as InfLLM[1] should be incorporated.

[1] InfLLM: Unveiling the Intrinsic Capacity of LLMs for Understanding Extremely Long Sequences with Training-Free Memory

**Questions:**

The author posits that “CLMs have seen samples similar to the current input during large-scale pre-training, their gradients gm, gm+1 tend to be zero”. Does this assumption potentially compromise the model's generalization ability in in-domain scenarios or with more recent data? More discussions are expected.

---

> ### Author Response · Authors · 2024-11-22
>
> **W1:** The baselines utilized in this work are relatively weak. More competitive baselines, such as InfLLM should be incorporated.
>
> **Response:** Please see our **response to General Weakness**, we have already included comparisons with InfLLM and several other baselines. The experimental results demonstrate that our method significantly outperforms the baselines.
>
> **Q1:** The author posits that "CLMs have seen samples similar to the current input during large-scale pre-training, their gradients gm, gm+1 tend to be zero". Does this assumption potentially compromise the model's generalization ability in in-domain scenarios or with more recent data? More discussions are expected.
>
> **Response:** This assumption is domain-independent and stems from the fact that LLMs have been exposed to vast and diverse data during pre-training. The assumption is not tied to any specific domain distribution and requires no in-domain fine-tuning, making it inherently generalizable.
>
> As demonstrated in Table 1, without any fine-tuning, our model achieves significant performance across multiple tasks, validating the effectiveness of this assumption. In our tests, we also explored incorporating the gradient terms from Equation (13) with weight coefficients rather than omitting them entirely. However, this modification showed no significant improvement compared to directly excluding the gradient terms.

---

> > ### Comment · Reviewer_7BNf · 2024-11-23
> >
> > Thank you for providing a detailed response with additional results.
> >
> > I still have a couple of questions and would appreciate further clarification:
> >
> > 1. Could you please elaborate on how the "Max Content Length set to 2048" was implemented across the different approaches for fair comparisons?
> >
> > 2. Could you provide a more in-depth analysis of why Stated-CLM performs poorly on code-related tasks? LLM's reasoning capability is greatly reflected by its ability to handle mathematical and coding problems. Does the result show that memory retrieval-based methods (infLLM) are more competitive than memory compression-based methods (Stated-CLM) on those more challenging tasks requiring long-term reasoning abilities?
> >
> > Besides, I recommend revising the paper with these additional baselines and discussions.

---

> ### Author Response · Authors · 2024-11-23
>
> **Response to Additional Q1:** For a fair comparison, we keep a max context size of 2048 for all methods. For _Stated-CLM_ and InfLLM, we set sink size=32 and recent size=2016 to ensure cache size=2048. For InfLLM, we additionally allow up to 8 blocks of length 128 as external memory modules. For H2O and THINK, we directly use the settings from the THINK paper with cache size=2048.  LLMLingua2 is a prompt compression method, which uses compression rate rather than cache size. So we adopted an approximate compression ratio (4x).
>
> **Response to Additional Q2:** We found that code-related tasks present more challenges in both generation and evaluation than other tasks in LongBench. The models sometimes failed to accurately follow instructions, producing results completely unrelated to the prompt. For instance, when asked to complete code, the model might generate a new similar query or simply restate the original query. In such cases, LongBench's evaluation metric (edit similarity) becomes ineffective, fluctuating between 0-0.3, resulting in unstable Code evaluation results. Therefore, for LongBench, we recommend **focusing more on the performance of tasks other than code-related ones**.
>
> We will add these additional baselines and discussions in the revised version.

---

> ### Comment · Reviewer_7BNf · 2024-11-27
>
> Thank you very much for your responses.
>
> * While the evaluation of the Code task shows some instability, Stated-CLM consistently performs poorly in this task, as indicated by the results in Table 1 and the additional baselines in the General Response.
>
> * Stated-CLM demonstrates superior performance in the early topic retrieval task as shown in Table 2. However, it should be noted that other strong baselines with memory capabilities were not compared in this evaluation.
>
> In light of these points, I maintain the view that memory retrieval-based methods (e.g., infLLM) are likely to be more competitive than memory compression-based methods (e.g., Stated-CLM) for tasks that require detailed and precise context information, such as coding, reasoning, "needle in a haystack" problems, etc.

---

> > ### Author Response · Authors · 2024-11-27
> >
> > **Response:** Memory-retrieval-based methods (like InfLLM) has advantages in "needle in a haystack"-style retrieval tasks, as they excel at retrieving specific segments. However, these methods become less effective when tasks require comprehension of the full context. We tested InfLLM on the early topic retrieval task as in Table 2, which demands understanding of global paragraph semantics and their orders. The results are shown below. As demonstrated, Stated-CLM shows significant performance advantages.
> >
> > | Model | LLaMA3-8B | LLaMA3.1-8B |
> > |-------|-----------|-------------|
> > | InfLLM | 44.00 | 53.33 |
> > | Stated-CLM | **64.00** | **76.00** |

---

### Official Review · Reviewer_coDH · 2024-10-31

**Soundness:** 2
**Presentation:** 3
**Contribution:** 2
**Rating:** 3
**Confidence:** 5

**Summary:**

Stated-CLM, by compressing adjacent tokens, retains contextual information to enhance memory and supports long contexts without the need for additional training. Extensive experiments have demonstrated the advantages of Stated-CLM relative to baselines.

**Strengths:**

1. The writing is good.
2. The article validates the advantages of Stated-CLM over multiple baselines, providing extensive experimental results.
3. The author provides relatively detailed mathematical derivations for the proposed method.

**Weaknesses:**

1. This paper’s main weakness, in my view: The paper emphasizes its benefits in "**CONTEXT MEMORIZATION**," but the compared baselines (StreamingLLM and LongCache) lack memory capabilities, so the experimental results are bound to be better. There are many memory-capable methods that support long contexts without requiring training, such as THINK [1], StreamingDialogue [2], and SampleAttention [3]. The paper lacks both a performance comparison with these types of methods and a comparison of space and time complexity.
2. Stated-CLM seems to struggle with very long inputs. Assuming the context window is 2,048 tokens and the input is 10,000 tokens, it would require an initial stated-CLM run to compress the tokens down to 5,000, then another compression to 2,500, and this process would need to be repeated until it is reduced to within 2,048. This is costly and cumbersome.
3. Stated-CLM does not support streaming generation. Section 3.4 indicates that this method must pause after generating a certain number of tokens to perform a stated-CLM step.


**Reference**

[1] THINK: THINNER KEY CACHE BY QUERY-DRIVEN PRUNING

[2] StreamingDialogue: Prolonged Dialogue Learning via Long Context Compression with Minimal Losses

[3] SampleAttention: Near-Lossless Acceleration of Long Context LLM Inference with Adaptive Structured Sparse Attention

**Questions:**

More comparisons are needed to evaluate the effectiveness of stated-CLM.

---

> ### Author Response · Authors · 2024-11-22
>
> **W1:** Compare with THINK, StreamingDialogue, and SampleAttention.
>
> **Response:** Please see our **response to General Weakness**. We have already included comparisons with THINK and several other baselines. The experimental results demonstrate that our method significantly outperforms the baselines.
>
> **W2 & W3:** Stated-CLM seems to struggle with very long inputs. Assuming the context window is 2,048 tokens and the input is 10,000 tokens, it would require an initial stated-CLM run to compress the tokens down to 5,000, then another compression to 2,500. Section 3.4 indicates that this method must pause after generating a certain number of tokens to perform a stated-CLM step.
>
> **Response:** We need to kindly point out that there is a misunderstanding of our generation process. We do not perform sequential compression from 10,000 tokens to 5,000 tokens to 2,500 tokens. As described in Section 3.4, our generation is streaming-based, where we compress chunk-size k tokens in a single operation each time k new tokens are generated. Compressing **k tokens** takes approximately the same time as **one backpropagation pass**.
>
> Firstly, to compress one token, the complexity of computing the optimal compression encoding e^* matches that of language model backpropagation. In Equation 13, we approximate the Hessian using the Fisher Information Matrix, which requires only one backpropagation pass to compute gradients (as shown in Equation 14).
>
> Secondly, for the chunking-wise compression described in Section 3.4, we only need **one** backpropagation pass to complete both the computation of e^* at all positions m (Equations 13 and 14) and the compression of k tokens. Consequently, the computational overhead introduced by compression during inference is minimal.
>
> We have empirically validated the inference efficiency of Stated-CLM in Figure 3.

---

> > ### Comment · Reviewer_coDH · 2024-11-25
> >
> > Thank you for your response. I still hold my opinion that the contribution of this paper is limited. I will raise the soundness score while maintaining my overall rating.

---

### Official Review · Reviewer_nf25 · 2024-11-01

**Soundness:** 3
**Presentation:** 2
**Contribution:** 3
**Rating:** 6
**Confidence:** 4

**Summary:**

The work explores the problem of long-range language modeling. The key idea is to compress adjacent pairs of tokens into single tokens, by which both the length of context and the information loss from compression are controlled. The authors derived a principle approach to estimate the tokens as compression of adjacent tokens. Experiments are conducted on LongBench and L-Eval with promising results.

To the best of my knowledge, the proposed method is new in the research of long-range modeling. The empricial results, in terms of both efficacy (e.g., Table 1) and efficiency (e.g., Figure 3) are significant. In addition to these ``pros'', I have concerns as follows:

1. Though the authors provide empricial studies to show the advantage of the method in terms of efficiency, I think a formal and rigid complexity analysis is needed, as the method needs complicated extra computation in each layer and for each input sequence (correct me if I am wrong).  Without such analysis, it is confusing why the method can scale linearly with respect to context length in inference (cf., Figure 3).

2. When do you perform compression? In each step of pre-training? or just in inference? More implementation details are needed.

3. The method is also related to LLMLingua (https://github.com/microsoft/LLMLingua) and Retrieval Pre-trained Transformer (https://arxiv.org/abs/2306.13421). The results could be more solid if these methods are involved in comparsion.

4. Proof-reading is needed. Typos like "manuplate" in line 153 and grammar errors like "we the..." in line 197 are annoying.

**Strengths:**

1. New approach to long-range language modelling
2. Impressive empirical results on benchmarks.

**Weaknesses:**

1. Some necessary details are missing.
2. Typos and grammar errors are annoying.

**Questions:**

1. Is the assumption of "cross parameter interactions are negligible (another typo in the draft...)" too strong for parameter estimation?

---

> ### Author Response · Authors · 2024-11-22
>
> **W1:** A formal and rigid complexity analysis is needed.
>
> **Response:** We would like to clarify that each compression takes approximately the same time as **one backpropagation pass**.
>
> Firstly, to compress two adjacent tokens, the complexity of computing the optimal compression encoding $e^*$ matches that of language model backpropagation. In Equation 13, we approximate the Hessian using the Fisher Information Matrix, which requires only one backpropagation pass to compute gradients (as shown in Equation 14).
>
> Secondly, for the chunking-wise compression described in Section 3.4, we only need **one** backpropagation pass to complete both the computation of $e^*$ at all positions $m$ (Equations 13 and 14) and the compression of chunk-size tokens. Consequently, the computational overhead introduced by compression during inference is minimal.
>
> We have empirically validated the efficient inference in Figure 3.
>
>
> **W2:** When do you perform compression? In each step of pre-training? or just in inference?
>
> **Response:** As stated in lines 68-69, Stated-CLM only compresses its states during inference, requiring no model training or fine-tuning. For more details of the chunking-wise compression, please refer to the response to W1.
>
> **W3:** The method is also related to LLMLingua and Retrieval Pre-trained Transformer. The results could be more solid if these methods are involved in comparison.
>
> **Response:** Please see our **response to General Weakness**, we have already included comparisons with LLMLingua2 and several other baselines. The experimental results demonstrate that our method significantly outperforms the baselines.
>
> **Q1:** Is the assumption of "cross parameter interactions are negligible" too strong for parameter estimation?
>
> **Response:** To validate this assumption, we compared the average values of two cases for LLaMA3-8B-instruct and LLaMA3.1-8B-instruct using the ShareGPT dataset:
> 1. Values in the diagonal of the Hessian matrix ($avg(|H_{ii}|)$)
> 2. Other values ($avg(|H_{ij}|)$, $i \neq j$)
>
> | Model | $avg(\|H_{ii}\|)$ | $avg(\|H_{ij,i\neq j}\|)$) |
> |-------|-----|-----|
> | LLaMA3-8B-instruct | 1.07E-08 | 3.50E-10 |
> | LLaMA3.1-8B-instruct | 5.68E-09 | 2.32E-10 |
>
> As shown in the table above, the off-diagonal elements are significantly smaller than the diagonal elements, supporting the validity of ignoring cross parameter interactions.

---

> > ### Comment · Reviewer_nf25 · 2024-11-25
> > **Thanks for the response**
> >
> > Thanks for the explanation. So with respect to the complexity analysis, is the overall complexity equivalent to that of the vanilla Transformer, or is it an order or even several orders of magnitude slower?

---

> > > ### Author Response · Authors · 2024-11-26
> > >
> > > The overall complexity is equivalent to that of a vanilla Transformer with KV cache. Empirical results are shown in Figure 3. StreamingLLM can be viewed as applying the vanilla Transformer with KV cache. For every $c$ steps (where $c$ is the chunk size), only one step requires one additional backpropagation pass, while the remaining $c-1$ steps maintain identical computational costs to the vanilla Transformer.

---

> > > > ### Comment · Reviewer_nf25 · 2024-11-26
> > > > **Thanks for the response**
> > > >
> > > > To some extent, the update addressed my concerns. Therefore, I slightly upgrade my rating.

---

### Official Review · Reviewer_LS35 · 2024-11-04

**Soundness:** 3
**Presentation:** 3
**Contribution:** 3
**Rating:** 5
**Confidence:** 4

**Summary:**

The paper introduces "Stated Causal Language Modeling" (stated-CLM), a novel approach to enhance the memory capacity of large language models (LLMs) without modifying their architectures or parameters. The method aims to address limitations in LLMs' context length and computational complexity by compressing rather than discarding low-weight tokens. Using techniques inspired by network pruning, stated-CLM compresses adjacent tokens into a single token, preserving critical context information while maintaining memory efficiency.

**Strengths:**

1. Innovative Approach: The proposed token compression approach enhances memory without architecture changes, a clear advantage over existing context segmentation methods.
2. Efficiency in Context Compression: Stated-CLM reduces memory usage while preserving essential information, enabling longer context retention compared to traditional methods.
3. Strong Performance: Experimental results on multiple benchmarks, including LongBench and TopicRet, show a marked improvement over baselines in memory and retrieval tasks.
4. Applicability Across Models: The method was tested on LLaMA, Mistral, and Gemma architectures, illustrating versatility across different LLMs.

**Weaknesses:**

1. One potential weakness is the paper’s limited exploration of the trade-offs between compression rates and downstream performance across a wider variety of tasks. While the LongBench and TopicRet benchmarks cover important aspects of long-context retention, it would strengthen the paper to evaluate stated-CLM on additional tasks that may reveal specific limitations in high-compression settings, such as more complex QA or summarization tasks.
2. The paper briefly mentions that low-weight tokens can still affect prediction outcomes, yet it does not quantitatively analyze the impact of compressing such tokens on information loss. To make this assessment more actionable, the paper could add metrics that measure the fidelity of compressed representations relative to uncompressed sequences. Additionally, examining token dependencies—particularly in sequences where tokens have subtle but cumulative importance—would reveal potential blind spots in the compression approach.

**Questions:**

1. How does stated-CLM handle information loss from compressed tokens, especially for tasks with subtle context dependencies? Can the authors provide more insight into the impact of compressing low-weight tokens on prediction accuracy?
2. What factors influenced the selection of specific hyperparameters (e.g., sink length and chunk length), and how adaptable are these parameters? It would be useful to understand why specific values were chosen for sink length, chunk length, and other hyperparameters, as well as any experiments the authors conducted to tune these parameters.

---

> ### Author Response · Authors · 2024-11-24
>
> **W1:** The paper's limited exploration of the trade-offs between compression rates and downstream performance across a wider variety of tasks.
>
> **Response:** We have comprehensively illustrated the relationship between compression rates and downstream performance **across six different LLMs with the representative HotpotQA task** in Figure 2. Our experimental results demonstrate that most models can maintain performance levels close to their full-context capabilities even when compressed to just 10% of the original context length.
>
>
>
> **W2&Q1:** It does not quantitatively analyze the impact of compressing such tokens on information loss. How does stated-CLM handle information loss from compressed tokens, especially for tasks with subtle context dependencies?
>
> **Response:** LLMs' predictive capability is viewed as their ability to compress massive text data. Based on this, the corresponding causal language modeling loss is a key metric for measuring language model performance. We analyzed loss changes under different compression ratios to reveal the impact of Stated-CLM's state compression on LLMs. Using ShareGPT samples, which contain high-quality, diverse, and challenging QA data, we randomly selected 100 samples with lengths between 8,000-12,000 tokens. The results are shown below:
>
> | LLaMA3-8B-Instruct | 2% | 10% | 20% | 30% | 40% | 100% |
> |-------------------|-----|-----|-----|-----|-----|------|
> | LongCache | 1.63 | 1.43 | 1.38 | 1.36 | 1.35 | - |
> | StreamingLLM | 1.65 | 1.44 | 1.38 | 1.36 | 1.35 | - |
> | Stated-CLM | **1.62** | **1.42** | **1.37** | **1.35** | **1.34** | 1.29 |
>
> As shown, while all baselines demonstrate compression capabilities, Stated-CLM achieves the best compression performance.
>
> **Q2:** What factors influenced the selection of specific hyperparameters (e.g., sink length and chunk length), and how adaptable are these parameters?
>
> **Response:** We evaluated the impact of sink length and chunk length using the multifieldqa(en) task from LongBench as our evaluation set.
>
> | Sink Length | Performance |
> |-------------|-------------|
> | 16 | 45.71 |
> | 32 | 45.13 |
> | 64 | 43.82 |
>
> | Chunk Length | Performance |
> |--------------|-------------|
> | 128 | 41.35 |
> | 256 | 39.24 |
> | 512 | 45.13 |
> | 1024 | 47.08 |
>
> The results show that increasing sink size has no significant impact on performance. This aligns with the attention sink phenomenon, where attention tends to concentrate on the initial tokens.
>
> Conversely, larger chunk lengths improve model performance. This is because new chunks are used to compute CLM loss (Equation (6)) for existing compressed tokens. Larger chunk sizes provide more informative CLM loss, better guiding token compression. However, longer chunk lengths require more GPU memory during CLM loss computation. Therefore, we chose chunk size=512 as a balanced setting.

---

> > ### Comment · Reviewer_LS35 · 2024-11-27
> >
> > Thank you for your rebuttal and your clarification for my concerns. I think the results are closed and not very distinct varying different compression ratios among different models. Can you explain the reason?

---

> > > ### Author Response · Authors · 2024-11-27
> > >
> > > Model performance is highly sensitive to next token prediction loss. Even small reductions in loss can lead to significant improvements in model capabilities.
> > >
> > > For reference, according to the LLaMa 3 paper, increasing training FLOPs from $1 \times 10^{21}$ to $1 \times 10^{22}$ only reduced the next token prediction loss on ARC Challenge from 1.35 to 1.32. Further increasing training FLOPs to $3.8 \times 10^{25}$ only brought the loss down to approximately 1.20.
> > >
> > > Therefore, the consistently lower loss demonstrated by Stated-CLM sufficiently validates its performance advantages.

---

### Author Response · Authors · 2024-11-22
**General Response**

**General Weakness:** More baselines are needed.

**Response:** We have supplemented several representative methods as baselines, including LLMLingua2, H2O, InfLLM, and THINK. The results on LongBench using LLaMA3-8B are shown in the table below, with Max Content Length set to 2048. The results of THINK and H2O are from [1], while other baseline implementations are from [2]. From the results, Stated-CLM shows significant improvements over all baselines.

[1] ThinK: Thinner Key Cache by Query-Driven Pruning

[2] https://github.com/henryzhongsc/longctx_bench

| Model | Single-DocQA | Multi-DocQA | Sum | Few-shot Learning | Synthetic | Code | Avg. |
|--------|--------------|--------------|------|-----------------|-----------|------|-------|
| LLMLlingua2-4x | 26.50 | 30.80 | 24.10 | 39.30 | 22.50 | 32.20 | 29.90 |
| H2O | 30.65 | 32.77 | 24.61 | 61.83 | 37.08 | 54.87 | 39.59 |
| InfLLM | 33.67 | 31.79 | 26.30 | **68.27** | 31.94 | 58.99 | 41.37 |
| Think(0.4)+H2O | 30.35 | 31.76 | 24.49 | 61.69 | 37.39 | **60.55** | 40.05 |
| **Stated-CLM** | **35.53** | **33.43** | **26.67** | 67.64 | **39.79** | 52.05 | **42.09** |

---

> ### Author Response · Authors · 2024-11-24
> **Implementation details**
>
> For a fair comparison, we keep a max context size of 2048 for all methods. For _Stated-CLM_ and InfLLM, we set sink size=32 and recent size=2016 to ensure cache size=2048. For InfLLM, we additionally allow up to 8 blocks of length 128 as external memory modules. For H2O and THINK, we directly use the settings from the THINK paper with cache size=2048.  LLMLingua2 is a prompt compression method, which uses compression rate rather than cache size. So we adopted an approximate compression ratio (4x).

---

### Meta-Review · Area_Chair_uxkX · 2024-12-18

**Metareview:**

The paper addresses the problem of long-range language modeling by compressing adjacent pairs of tokens into single tokens.

The promplem address is important, this paper is well-motivated and easy-to-read.

However, some reviewers raised critical concerns such as comparisons with improper baselines (models whose context memorization capability is not strong), limited contributions despite the additional experiments during the rebuttal.

By AC-reviewer discussion, the reviewers agree that this paper is not sufficient for ICLR quality.

So, AC recommends rejecting this paper.

**Additional Comments On Reviewer Discussion:**

The initial scores were 5, 5, 3, 5.

During the rebuttal period, only one reveiwer (nf25) raised his/her score to borderline accept.

So, the final scores are 5, 6, 3, 5.

For the AC-reviewer discussion, nf25 also agreed to reject this paper due to limited contributions following other reviewers opinions.

---

### Decision · Program_Chairs · 2025-01-22

Reject